# A randomised controlled trial of a facilitated home-based rehabilitation intervention in patients with heart failure with preserved ejection fraction and their caregivers: the REACH-HFpEF Pilot Study

Chim C Lang,[1] Karen Smith,[1,2] Jennifer Wingham,[3,4] Victoria Eyre,[5] Colin J Greaves,[3] Fiona C Warren,[3] Colin Green,[3] Kate Jolly,[6] Russell C Davis,[7] Patrick Joseph Doherty,[8] Jackie Miles,[9] Nicky Britten,[3] Charles Abraham,[3] Robin Van Lingen,[10] Sally J Singh,[11] Kevin Paul,[12] Melvyn Hillsdon,[13] Susannah Sadler,[3] Christopher Hayward,[14] Hayes M Dalal,[3,4] Rod S Taylor,[3] and on behalf of the REACH-HF investigators,

For numbered affiliations see end of article.

**Correspondence to**
Professor Chim C Lang;
c.c.lang@dundee.ac.uk

## ABSTRACT

**Introduction** Home-based cardiac rehabilitation may overcome suboptimal rates of participation. The overarching aim of this study was to assess the feasibility and acceptability of the novel Rehabilitation EnAblement in CHronic Hear Failure (REACH-HF) rehabilitation intervention for patients with heart failure with preserved ejection fraction (HFpEF) and their caregivers.

**Methods and results** Patients were randomised 1:1 to REACH-HF intervention plus usual care (intervention group) or usual care alone (control group). REACH-HF is a home-based comprehensive self-management rehabilitation programme that comprises patient and carer manuals with supplementary tools, delivered by trained healthcare facilitators over a 12 week period. Patient outcomes were collected by blinded assessors at baseline, 3 months and 6 months postrandomisation and included health-related quality of life (primary) and psychological well-being, exercise capacity, physical activity and HF-related hospitalisation (secondary). Outcomes were also collected in caregivers. We enrolled 50 symptomatic patients with HF from Tayside, Scotland with a left ventricular ejection fraction ≥45% (mean age 73.9 years, 54% female, 100% white British) and 21 caregivers. Study retention (90%) and intervention uptake (92%) were excellent. At 6 months, data from 45 patients showed a potential direction of effect in favour of the intervention group, including the primary outcome of Minnesota Living with Heart Failure Questionnaire total score (between-group mean difference −11.5, 95% CI −22.8 to 0.3). A total of 11 (4 intervention, 7 control) patients experienced a hospital admission over the 6 months of follow-up with 4 (control patients) of these admissions being HF-related. Improvements were seen in a number intervention caregivers' mental health and burden compared with control.

### Strengths and limitations of this study

► Rehabilitation EnAblement in CHronic Heart Failure (REACH-HF) is the first comprehensive home-based, self-management cardiac rehabilitation intervention for patients with heart failure with preserved ejection fraction (HFpEF) and their caregivers.
► The findings of this pilot study support the feasibility and acceptability of the home-based REACH-HF rehabilitation intervention in patients with HFpEF and their caregivers and indicate that it is feasible to recruit and retain participants in a randomised trial with follow-up.
► Potential favourable impacts of the REACH-HF intervention on caregiver mental health and measures of burden were observed in this pilot study.
► This study was not designed or powered to definitively assess the efficacy or safety of the REACH-HF intervention in HFpEF.
► Generalisability of this study's findings is limited, given it was based in a single centre.

**Conclusions** Our findings support the feasibility and rationale for delivering the REACH-HF facilitated home-based rehabilitation intervention for patients with HFpEF and their caregivers and progression to a full multicentre randomised clinical trial to test its clinical effectiveness and cost-effectiveness.

**Trial registration number** ISRCTN78539530.

## INTRODUCTION

Epidemiological data show that approximately half of those patients with clinical features of heart failure (HF) have preserved

**BMJ**

ejection fraction (HFpEF).[1] In contrast to HF with reduced ejection fraction (HFrEF), the prevalence of HFpEF is increasing.[2] Importantly, the substantial burden from HFpEF appears to be similar to HFrEF, measured by exercise intolerance, poor health-related quality of life (HRQoL), mortality, increased hospital admissions and higher healthcare costs.[3] Although drug and device therapy have helped to improve outcomes in HFrEF, prognosis in HFpEF remains unchanged, with no large-scale randomised trial demonstrating significant treatment benefits that alter the natural course of HFpEF or lower mortality.[4 5] However, systematic reviews and meta-analyses have shown promising evidence for the benefit of exercise-based cardiac rehabilitation (CR) in HFpEF.[6 7] A recent meta-analysis of eight randomised trials in 317 patients with HFpEF found exercise-based CR significantly improved exercise capacity and HRQoL compared with usual care.[7] The CR programmes undertaken in these trials were predominantly group-based, supervised and delivered in centre-based settings.

Participation of patients with HF in CR remains suboptimal.[8 9] A UK survey found that only 16% of CR centres provided an HF programme; commonly cited reasons for the lack of CR provision were a lack of resources and exclusion from commissioning agreements.[9] Two main reasons given by patients for failing to take part in CR are difficulties with regular attendance at their local hospital centre and reluctance to join group-based classes.[9]

There is increasing recognition of the possibility of alternative delivery models of CR, such as home-based programmes, in order to overcome suboptimal rates of CR uptake seen with HF.[10 11] Facilitated home-based CR has been shown to provide similar benefits to centre-based CR in terms of clinical and HRQoL outcomes at equivalent cost for those with HF and following myocardial infarction and revascularisation.[11 12]

The Rehabilitation EnAblement in CHronic Heart Failure (REACH-HF) programme of research was designed to develop and evaluate a home-based comprehensive self-management rehabilitation intervention, including a self-care manual, an exercise programme, and facilitation by health professionals designed to improve self-management and HRQoL in people with HF.[13 14] In addition to REACH-HF, the intervention includes a 'Family and Friends Resource' designed to support caregivers.

The overarching aim of this study was to assess the feasibility of undertaking a definitive randomised trial to assess the clinical effectiveness and cost-effectiveness of the REACH-HF intervention in patients with HFpEF and their caregivers. Specific objectives of the study were to: (1) Assess the acceptability of the study design and procedures to participants (patients and caregivers). (2) Assess feasibility and experience of the delivery of intervention for participants and healthcare professional facilitators. (3) Identify barriers to participation in the intervention and study procedures. (4) Inform a definitive study sample size. (5) Assess methods for the collection of data including resource use and costs. (6) Assess the fidelity of the delivery of the REACH-HF intervention by healthcare professional facilitators.

## METHODS

The study design and methods have been described in the published study protocol.[14] The study is reported in accordance with the Consolidated Standards of Reporting Trials (CONSORT) extension for pilot trials.[15]

### Design

The REACH-HFpEF pilot study was a single-centre (Tayside, Scotland) two-group randomised controlled trial with parallel mixed-methods feasibility evaluation and assessment of costs. Participants were individually randomised in a 1:1 ratio to the REACH-HF intervention plus usual care (intervention group) or usual care alone (control group). Given the nature of the REACH-HF intervention, it was not possible to blind participants or those involved in the provision of care. However, the statistician (FCW) undertaking the data analysis was blinded to treatment allocation and we also blinded researchers undertaking collection of outcome data to minimise potential bias. We assessed the fidelity of blinding by asking outcome assessors at each follow-up visit to guess patient group allocation. Unblinding of groups did not take place until after data analysis and the blinded results had been presented to the Trial Management Group and interpretation of results was agreed.

### Study population

The study population included patients and their caregivers. Participating patients were aged 18 years or older and had a confirmed diagnosis of HFpEF on echocardiography, radionuclide ventriculography or angiography (ie, left ventricular ejection fraction ≥45% within the last 6 months prior to randomisation). Patients who had undertaken CR within 6 months prior to enrolment were excluded, as were patients with a contraindication to exercise testing or exercise training (with consideration of adapted European Society of Cardiology guidelines for HF).[14 16] Participating caregivers were aged 18 years or older and provided unpaid support to participating. Patients who did not have an identified caregiver were able to participate, as were those whose caregiver was not willing to participate in the study.

### Intervention

The REACH-HF intervention is described in detail elsewhere.[17] In summary, REACH-HF is a comprehensive self-management programme informed by evidence, theory and service user perspective. It comprises the 'Heart Failure Manual' (REACH-HF manual), relaxation compact disc (CD), chair-based exercise digital versatile disc (DVD), a 'Progress Tracker' tool for patients and a 'Family and Friends Resource' for caregivers. Participating patients and caregivers worked through the REACH-HF manual over a 12-week period with facilitation

by two trained cardiac nurses. The facilitators provided support as needed of which at least one was face to face and two were by telephone contacts. The REACH-HF manual incorporates five core informative and interactive elements covering a wide range of topics relating to living with/adapting to living with HF, and includes:

1. A progressive exercise training programme, tailored according to initial fitness assessments, delivered as a walking programme or a chair-based exercise DVD, or a combination of the two (as selected by the patient).
2. Managing stress/breathlessness/anxiety.
3. HF symptom monitoring.
4. Taking medication.
5. Understanding HF (and why self-management helps).

The REACH-HF manual was designed for patients with HFrEF (in terms of coverage of medication and explanations of condition). There was limited evidence to guide the development of the REACH-HF manual for patients with HFpEF. Thus it was adapted for this pilot study to allow evaluation in patients with HFpEF. The majority of the self-management advice in all other sections of the REACH-HF manual is relevant to all patients with HF and corresponds to national HF guidelines.[18 19] The core priorities for caregiver elements of the intervention were:

1. To facilitate improvement in patient HRQoL by helping them to achieve the core priorities for change.
2. To improve HRQoL for caregivers by acting to maintain their own health and well-being.

### Usual care

Both intervention and control group patients received usual medical management for HF according to current guidelines.[18 19]

### Outcome measures and follow-up

We collected the following pilot study outcomes: recruitment rate for participants (patients and caregivers) across the various recruitment pathways; attrition and loss to follow-up; completeness of participant outcome measures at follow-up; fidelity of the REACH-HF manual delivery by intervention facilitators (sample of patient-facilitator contacts for a sample of six patients were audio recorded and independently reviewed using a 13-item checklist (developed by CJG and JW) by two researchers (KS and Karen Coyle)); acceptability of the intervention (via face-to-face semistructured interviews with a purposive sample of 15 patients, 7 caregivers and both facilitators at the end of the intervention delivery period); and acceptability of study participation to participants (via interviews and questionnaires).

The following participant outcomes proposed for a future definitive trial were collected at baseline (prerandomisation) and follow-up at 4 months and 6 months postrandomisation:

*Patients:* disease-specific HRQoL (Minnesota Living with Heart Failure Questionnaire (MLHFQ) (primary outcome)[20] and Heart Related Quality of Life (HeartQoL) Questionnaire);[21] clinical events (all-cause mortality, hospital admission related to HF and not related to HF (relatedness was independently adjudicated by a panel of three cardiologists)); exercise capacity (incremental shuttle walking test (ISWT));[22] physical activity level (GeneActive accelerometry over a 7-day period);[23] psychological well-being (Hospital Anxiety and Depression Scale Questionnaire, HADS);[24] generic HRQoL (EQ-5D-5L questionnaire);[25] Self-care of HF Index Questionnaire (SCHFI);[26] healthcare utilisation (primary and secondary care contacts, social care contacts and relevant medication usage, reported by patient participants); and safety outcomes (serious adverse events).

*Caregivers:* Caregiver Burden Questionnaire for HF (CBQ-HF);[27] Caregiver Contribution to SCHFI Questionnaire (CC-SCHFI);[26] Family Caregiver Quality of Life Scale Questionnaire;[28] Generic HRQoL (EQ-5D-5L);[25] and psychological well-being (HADS).[24]

### Data analysis

Our planned recruitment target of 50 patients allowed us to achieve the feasibility aims and objectives of this study, that is, an estimate of attrition, estimates of the SD of the primary and secondary outcomes to inform power for a future definitive trial, and sufficient numbers for qualitative interviews.

We report the mean and SD (or relevant summary statistics) for both groups for all patient and caregiver outcomes at each follow-up point and the mean (and 95% CIs) for the between-group difference in outcomes at 6-month follow-up using linear regression models adjusting for baseline outcome. Given the pilot nature of this trial, we do not report p values for the comparison of outcomes between groups. All analyses are based on the intention-to-treat principle (patients are analysed according to their original random allocation) using observed data only.

Data on patient resource use related to health and social care were collected using a standardised resource use questionnaire at baseline (for previous 6 months) and at 4-month and 6-month follow-ups. Unit costs per item of resource use were obtained from published estimates and where necessary inflated to 2016 prices using the Healthcare and Community Health Services index (see online supplementary eTable 1).[29] These unit costs were then applied to the resource use reported at patient level to estimate the delivery costs associated with the REACH-HF manual, and the total costs associated with health and social care at baseline and over the 6-month follow-up. As with clinical outcome, costs are presented descriptively. EQ-5D-3L utilities were obtained using existing crosswalk values from EQ-5D-5L.[30] All outcomes and costs analyses were conducted using Stata (V.14.2; StataCorp, College Station, Texas, USA). Patient, caregiver and facilitator interviews were transcribed verbatim and analysed using thematic analysis and will be fully reported elsewhere.[31]

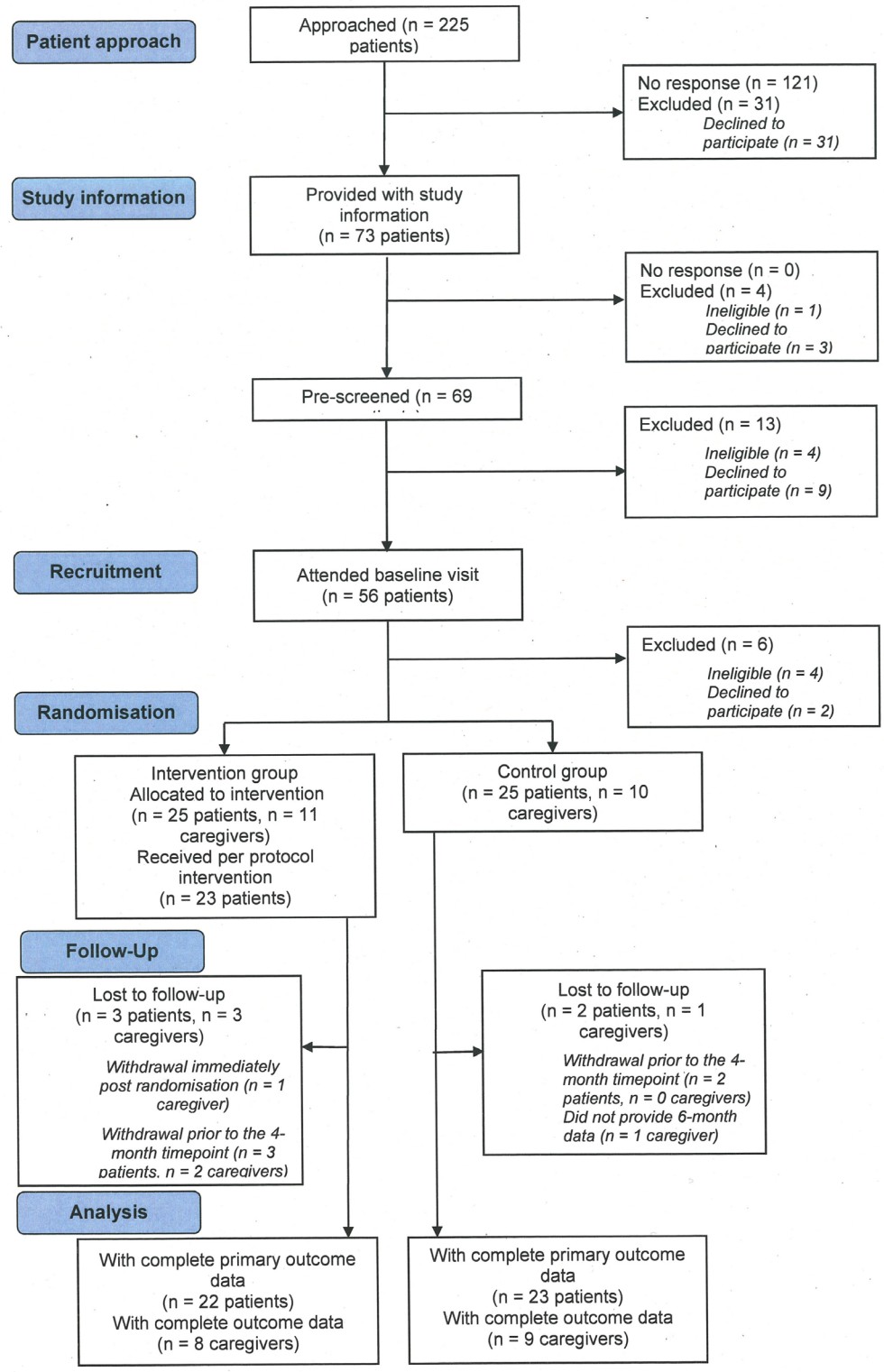

**Figure 1** Consolidated Standards of Reporting Trials (CONSORT) flow chart.

## RESULTS

### Recruitment and retention of patients and caregivers and acceptability of trial design

Study enrolment, allocation and follow-up of study participants are summarised in the CONSORT flow diagram shown in figure 1. Between April 2015 and June 2016, 225 potential patients were approached and 50 were randomised (intervention group 25; control group 25) that is, 22% (95% CI 17% to 28%) of patients approached. The original forecast was a recruitment rate of five patients per month. However, the actual recruitment rate during the trial was 4.5 patients per month, resulting in a 1-month extension to the period of recruitment. A caregiver was recruited in connection with 21 (42%) patient participants (intervention group 11; control group 10).

**Table 1** Baseline demographic characteristics

**(A) Patient characteristics**

| | Intervention n=25 mean (SD) or n (%) | Control n=25 mean (SD) or n (%) |
|---|---|---|
| Gender: male | 9 (36) | 14 (56) |
| Age (years) | 71.8 (9.9) | 76.0 (6.6) |
| BMI (kg$^2$/m) | 32.1 (6.3) | 32.2 (5.3) |
| Ethnic group: white | 25 (100) | 25 (100) |
| Relationship status: | | |
| Single | 4 (16) | 2 (8) |
| Married | 14 (56) | 8 (32) |
| Divorced/civil | 1 (4) | 3 (12) |
| Partnership dissolved/widowed | 6 (24) | 12 (48) |
| Domestic residence: | | |
| Live alone | 9 (36) | 14 (56) |
| Spouse/partner only | 14 (56) | 8 (32) |
| Spouse/partner and child > | 0 (0) | 2 (8) |
| 18 years other adult family members only | 2 (8) | 1 (4) |
| Smoking status | | |
| Never smoked | 2 (8) | 2 (8) |
| Ex-smoker | 15 (60) | 14 (56) |
| Current smoker | 8 (32) | 9 (36) |
| NYHA status | | |
| Class I | 1 (4) | 1 (4) |
| Class II | 15 (60) | 16 (64) |
| Class III | 9 (36) | 8 (32) |
| Class IV | 0 (0) | 0 (0) |
| Cause of heart failure* | | |
| Ischaemic | 8 (32) | 16 (64) |
| Non-ischaemic | 16 (64) | 8 (32) |
| Unknown | 1 (4) | 1 (4) |
| Number of comorbidities | | |
| 0 | 7 (28) | 12 (48) |
| 1 | 15 (60) | 6 (24) |
| 2 | 3 (12) | 4 (16) |
| 3 | 0 | 2 (8) |
| 4 | 0 | 1 (4) |
| Previous myocardial infarction | 4 (16) | 5 (20) |
| Previous atrial fibrillation/atrial flutter | 6 (24) | 13 (52) |
| Hypertension | 18 (72) | 14 (56) |
| Diabetes mellitus | 9 (36) | 6 (24) |
| Chronic renal impairment | 3 (13) | 10 (40) |
| Time since diagnosis of heart failure (years) | | |
| <1 | 6 (24) | 4 (16) |
| 1 to 2 | 7 (28) | 7 (28) |
| >2 | 12 (48) | 15 (60) |
| Medication | | |
| β-blocker | 18 (72) | 13 (52) |

Continued

**Table 1** Continued

**(A) Patient characteristics**

| | Intervention n=25 mean (SD) or n (%) | Control n=25 mean (SD) or n (%) |
|---|---|---|
| Angiotensin 2 receptor antagonist | 7 (28) | 7 (28) |
| ACE inhibitor | 11 (44) | 14 (56) |
| Main activity: | | |
| In employment or self-employment | 0 (0) | 1 (4) |
| Retired | 22 (88) | 24 (96) |
| Unemployed | 2 (8) | 0 (0) |
| Other | 1 (4) | 0 (0) |
| Education | | |
| Postschool | 7 (28) | 7 (28) |
| Degree | 5 (20) | 5 (20) |
| Pro-BNP levels | | |
| ≤2000 pg/mL | 23 (92) | 22 (88) |
| >2000 pg/mL | 2 (8) | 3 (12) |

**(B) Caregiver characteristics**

| | Intervention n=11† mean (SD) or n (%) | Control n=10 mean (SD) or n (%) |
|---|---|---|
| Gender: male | 3 (30) | 2 (20) |
| Age (years) | 59.3 (14.0) | 64.8 (11.6) |
| Relationship to patient | | |
| Partner | 4 (40) | 6 (60) |
| Son/daughter | 3 (30) | 4 (40) |
| Sibling | 2 (20) | 0 (0) |
| Friend | 1 (10) | 0 (0) |

*Cause of HF determined by principal investigator.
†One caregiver withdrew shortly after randomisation and did not provide baseline data.
BNP, B-tyep Natriuretic Peptide; NYHA, New York Heart Association.

At the 6-month follow-up, 5 out 50 (10%, 95% CI 3% to 22%) patients were lost to follow-up. Seventeen out of the 21 recruited caregivers provided follow-up data at 6 months.

Patients and caregivers rated a high level of satisfaction with their participation in the trial (see online supplementary eTable 2).

### Baseline characteristics of patients and caregivers

There was evidence of imbalance between intervention and control group patients in terms of their baseline demographic characteristics (see table 1). Compared with the control group, the intervention group included a higher proportion of women, and lower proportions of patients with an ischaemic diagnosis, with atrial flutter/atrial fibrillation, and with chronic renal failure; also, the intervention group had a younger mean age. Caregivers were typically the partner or children of patients, were of a younger mean age than participating patients and predominantly female.

**Table 2** Rehabilitation EnAblement in CHronic Heart Failure (REACH-HF) intervention delivery: healthcare resource use and costs

| | Number of patient contacts, mean (SD) | Duration of patient contacts' contact (min), mean (SD) | Duration of facilitator non-contact planning (min), mean (SD) | Duration of facilitator travel (min), mean (SD) |
|---|---|---|---|---|
| Face-to-face contacts/patient | 5.1 (1.5) | 60.6 (29.6) | 17.2 (24.4) | 40.2 (37.4) |
| Telephone contacts/patient | 1.1 (1.3) | 7.7 (4.0) | 8.0 (9.5) | |
| Total contacts/patient | 6.2 (1.6) | | | |
| Total time, face-to-face contacts | | 308.9 (123.3) | | |
| Total time, telephone contacts | | 8.8 (10.3) | | |
| Total facilitator planning/non-contact time, face-to-face, minutes | | 87.4 (55.8) | | |
| Total facilitator planning/non-contact time, telephone, minutes | | 9.1 (12.6) | | |
| Overall total time input, time | | 414.2 (145.4) | | |
| | | Cost per patient*, mean (SD) | | |
| Estimated total HF facilitator cost | | £303.64 (£106.59) | | |
| Other resource use/costs | | | | |
| Consumables (1 × manual) | | £25.00 | | |
| DVDs (×2, at £7.50 each) | | £15.00 | | |
| Distribution of HF facilitator training costs, per participant † | | £18.97 | | |
| Estimated total delivery cost of the REACH-HF intervention | | £362.61 | | |

*Unit costs—staff: Staff grade equivalent to 'Community Nurse' (includes district nursing sister, district nurse) and Nurse Specialist (community), from Curtis and Burns, Unit Costs of Health and Social Care 2016, p141-142. Based on Agenda for Change band 6 (staff salary at £32 114 per annum). Estimated cost per hour = £44 (Curtis and Burns, 2016); Includes salary, salary on costs, overheads (management costs and non-staff costs (including travel/transport)), capital overheads, and excludes costs for qualifications.
†Training cost per REACH-HF facilitator, specific to delivery of the REACH-HF intervention, are estimated at £1897 (involving 3 days, that is, 24 hours training at £44/hour; costs for trainer/s per trainee at £366, assuming eight trainees per 3-day course, and trainers at Agenda for Change, Band 8a, £61/hour (Curtis and Burns, 2016); cost for REACH-HF facilitator manual at £400 each; plus estimate of consumables for training sessions). These costs are distributed across the first 100 participants/patients receiving the intervention, resulting in an estimate of £18.97 per participant.

## Completion of outcome measures by patients and caregivers and fidelity of blinding by outcome assessors

We collected data from 45/50 patients (90%, 95% CI 78% to 97%) at the 6-month follow-up on MLHFQ, our proposed primary outcome. Levels of completion of patient secondary outcomes and caregiver outcomes were consistently high (≥76% of participants for all outcomes). The one exception was ISWT, which had notably lower level of completion (35 (78%) patients at the 4-month follow-up and 33 (73%) patients at the 6-month follow-up).

Outcome assessors correctly guessed patient group allocation in 22% of cases (10/45) at 4 months and 20% of cases (19/45) at 6 months, indicating that blinding was likely to have been maintained.

## Acceptability of patients, carers and facilitators of the REACH-HF intervention and fidelity of intervention delivery by facilitators

Qualitative interviews and observations of the patients' and caregivers' interactions with the facilitator indicated high levels of satisfaction, acceptability and the feasibility of delivering the REACH-HF intervention in patients with HFpEF (see online supplementary eTable 3). One of the most highly valued elements of REACH-HF by participants was the role of the facilitator, who was seen to act as an educator, a source of emotional support and reassurance as well as a motivator and enabler.

Of the six patients selected for inclusion, a total of ~45 hours of patient-facilitator interaction was used for analysis. Fidelity scoring indicated adequate delivery (defined as a score of 3 or more) for most aspects of the intervention by the two facilitators (see online supplementary eTable 4). Of the six patients selected for inclusion, a total of ~45 hours of patient-facilitator interaction was used for analysis. Mean score for items 9 (addressing emotional consequences of being a caregiver) and 11 (caregiver health and well-being) was less than 3.

**Table 3** Patient outcomes at baseline and follow-up

| | Baseline | | Four-month follow-up | | Six-month follow-up | | Mean between-group difference* (95% CI) |
|---|---|---|---|---|---|---|---|
| | Intervention mean (SD), N | Control mean (SD), N | Intervention mean (SD), N | Control, mean (SD), N | Intervention mean (SD), N | Control mean (SD), N | |
| **Primary outcome** | | | | | | | |
| MLHFQ, Overall | 38.2 (27.6), 25 | 36.0 (26.5), 25 | 35.5 (28.3), 22 | 37.8 (27.9), 23 | 29.2 (25.8), 22 | 38.7 (30.1), 23 | −11.5 (−22.8 to 0.3) |
| MLHFQ, Physical | 21.6 (13.4), 25 | 19.8 (12.4), 25 | 19.4 (13.5), 22 | 20.7 (12.8), 23 | 16.2 (12.3), 21 | 20.3 (13.6), 23 | −4.7 (−10.1 to 0.8) |
| MLHFQ, Emotional | 7.8 (9.1), 25 | 7.8 (8.4), 25 | 8.0 (8.5), 22 | 9.1 (8.6), 23 | 6.8 (8.1), 21 | 9.0 (8.5), 23 | −2.7 (−6.0 to 0.6) |
| **Secondary outcomes** | | | | | | | |
| HADS, Anxiety | 5.6 (4.8), 25 | 6.1 (4.9), 25 | 5.7 (4.8), 22 | 6.4 (5.4), 23 | 5.5 (5.1), 21 | 6.0 (5.1), 23 | −0.2 (−2.6 to 2.1) |
| HADS, Depression | 6.2 (4.2), 25 | 5.6 (4.1), 25 | 5.6 (4.4), 22 | 6.6 (4.5), 23 | 5.4 (4.3), 21 | 6.9 (5.2), 23 | −1.5 (−3.4 to 0.3) |
| Heart-QoL, Global | 1.4 (0.8), 25 | 1.6 (0.9), 25 | 1.5 (1.0), 22 | 1.4 (1.0), 23 | 1.8 (0.8), 21 | 1.4 (1.1), 23 | 0.5 (0.0 to 0.9) |
| Heart-QoL, Physical | 1.2 (0.8), 25 | 1.4 (1.0), 25 | 1.3 (1.0), 22 | 1.3 (1.0), 23 | 1.6 (0.8), 21 | 1.3 (1.1), 23 | 0.5 (0.0 to 1.0) |
| Heart-QoL, Emotional | 2.0 (1.0), 25 | 2.0 (1.0), 25 | 2.0 (1.0), 22 | 1.9 (1.0), 23 | 2.2 (1.0), 21 | 1.8 (1.1), 23 | 0.3 (−0.1 to 0.8) |
| EQ-5D-3L, Index Score | 0.57 (0.29), 25 | 0.58 (0.31), 24 | 0.60 (0.28), 22 | 0.52 (0.34), 23 | 0.65 (0.31), 21 | 0.55 (0.29), 23 | 0.11 (−0.04 to 0.26) |
| SCHFI, Maintenance | 51.9 (13.9), 25 | 45.3 (16.5), 25 | 68.9 (14.9), 22 | 49.6 (14.4), 23 | 64.2 (12.8), 21 | 48.9 (14.3), 23 | 9.9 (2.5 to 17.3) |
| SCHFI, Management | 37.6 (20.7), 23 | 37.8 (18.4), 18 | 48.9 (26.5), 19 | 32.6 (19.2), 17 | 45.0 (2.7), 14 | 37.6 (23.5) 15 | 8.0 (−8.9 to 25.0) |
| SCHFI, Confidence | 60.4 (25.5), 25 | 56.9 (23.0), 25 | 65.2 (18.7), 22 | 49.5 (24.9), 23 | 62.1 (20.0), 21 | 53.4 (26.1), 23 | 6.6 (−6.7 to 19.9) |
| ISWT (metres) | 183.6 (174.2), 25 | 157.6 (117.8), 23 | 218.9 (185.5), 18 | 178.2 (115.0), 17 | 224.7 (161.4), 17 | 183.8 (98.1), 16 | −2.1 (−39.4 to 35.2) |
| Accelerometry, number of days/week with at least 10minutes/day activity>100mg | 5.8 (2.3), 25 | 5.9 (2.0), 25 | 5.6 (2.4), 21 | 5.7 (1.9), 21 | 4.9 (2.7), 19 | 6.0 (2.1), 20 | −0.4 (−1.3 to 0.5) |
| Accelerometry, average time/day at ≤20mg (mins) | 1126 (98), 25 | 1090 (112), 25 | 1115 (110), 21 | 1103 (124), 21 | 1136 (101), 19 | 1098 (114), 20 | −10 (−49 to 28) |
| Accelerometry, average time/day at 21 mg to 40mg (mins) | 128 (33), 25 | 152 (39), 25 | 140 (38), 21 | 143 (36), 21 | 134 (37), 19 | 148 (41), 20 | 12 (−4 to 29) |
| Accelerometry, average time/day at 41 mg to 60mg (mins) | 77 (27), 25 | 87 (29), 25 | 79 (29), 21 | 84 (33), 21 | 75 (25), 19 | 85 (27), 20 | 1 (−10 to 12) |
| Accelerometry, average time/day at 61 mg to 80mg (mins) | 45 (20), 25 | 47 (20), 25 | 45 (23), 21 | 45 (21), 21 | 40 (20), 19 | 45 (19), 20 | −1 (−9 to 6) |
| Accelerometry, average time/day at 81 mg to 100mg (mins) | 25 (14), 25 | 25 (15), 25 | 25 (15), 21 | 25 (17), 21 | 22 (15), 19 | 25 (15), 20 | −1 (−6 to 4) |
| Accelerometry, average time/day at >100mg (mins) | 39 (30), 25 | 40 (48), 25 | 36 (31), 21 | 39 (52), 21 | 32 (30), 19 | 39 (48), 20 | −2 (−9 to 5) |

*Mean between-group differences (intervention minus control) adjusted for baseline values.
HADS, Hospital Anxiety and Depression Scale; Heart-QoL, heart-related quality of life; ISWT, incremental shuttle walking test; MLHFQ, Minnesota Living with Heart Failure Questionnaire; SCHFI, Self-care of Heart Failure Index.

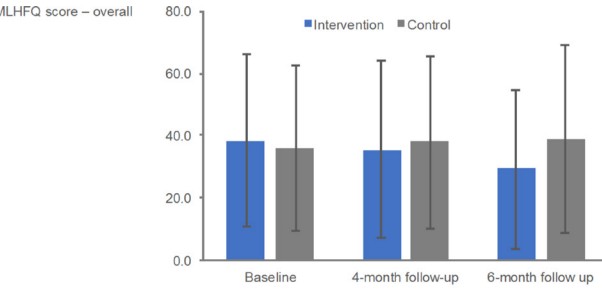

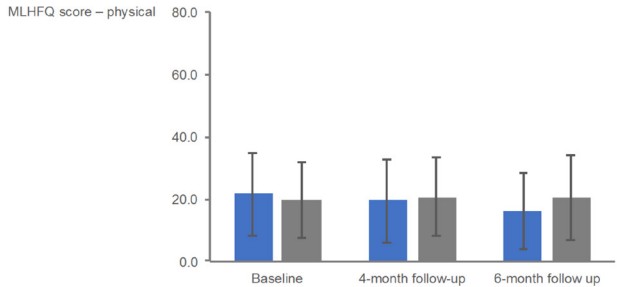

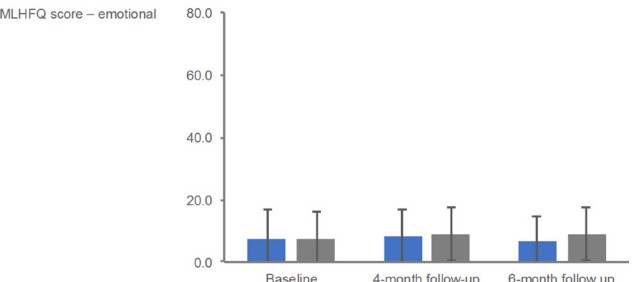

**Figure 2** Minnesota Living With Heart failure Questionnaire (MLHFQ) outcomes at baseline and at 4-month and 6-month follow-ups.

### Patient adherence to the REACH-HF intervention

Twenty-three of the 25 (92%) intervention patients met our minimum adherence criteria of attendance, that is, attendance at the first face-to-face meeting with the facilitator and at least two further facilitator contacts (either face-to-face or telephone). In these patients, the mean number of facilitator contacts was 6.2 (SD 1.6), the majority of which was face-to-face contacts (mean 5.1; SD 1.5) and the remainder was telephone contacts (mean 1.1, SD 1.3) (see table 2).

### PARTICIPANT OUTCOMES
#### Patients

Patient outcome results at baseline, and 3-month and 6-month follow-ups, and between-group differences at the 6-month follow-up are shown in table 3 (see online supplementary eTable 5 for baseline follow-up within group changes). At 6 months, a number of patient outcomes potential direction of effect in favour of intervention, including Minnesota Living With Heart Failure Questionnaire (MLWHFQ) total score (figure 2)

(between-group mean difference: −11.5, 95% CI −22.8 to 0.3), HeartQoL Global Score (0.5, 95% CI 0.0 to 0.9), EQ-5D-3L Utility Index (0.11, 95% CI −0.04 to 0.26), HADS Depression Score (−1.5, 95% CI −3.4 to 0.3) and SCHFI Maintenance Score (9.5, 95% CI 2.5 to 17.3). The directions of possible intervention effects were less clear for the outcomes of ISWT and level of physical activity.

At the 6-month follow-up, 11 (4 intervention, 7 control) patients experienced a hospital admission with 4 (all control) of these admissions being HF-related. All these serious adverse events were considered to be unrelated to the study processes or to the REACH-HF intervention. One control patient died related to HF shortly after the 6-month follow-up.

### Caregivers

Caregiver outcome results at baseline and 3-month and 6-month follow-ups are shown in table 4 (see online supplementary eTable 6 for within-group results). There were indications of a favourable intervention effect for some outcomes including HADS and CBQ-HF emotional and CC-SCHFI maintenance domain scores.

### Healthcare utilisation and intervention costs

The average cost of the REACH-HF intervention per patient was estimated to be £362.61. The intervention cost breakdown is provided in table 2. The wider healthcare and societal utilisation and costs for intervention and control groups are summarised in online supplementary eTable 7.

### DISCUSSION

The findings of this pilot study support the feasibility and acceptability of the home-based REACH-HF rehabilitation intervention in patients with HFpEF and their caregivers, and indicate that it is feasible to recruit and retain participants in a randomised trial of 6 months follow-up. The intervention was well received by patients, caregivers and healthcare facilitators and intervention adherence was good. At follow-up, compared with controls, a number of patient outcomes showed a potential direction of effect in favour of the intervention group, including our proposed primary outcome of disease-specific HRQoL, MLWHFQ. We also saw potentially favourable impacts of the REACH-HF intervention on caregiver mental health and measures of burden.

The promising results of this study support the emerging evidence of the impact of exercise-based CR interventions in HFpEF.[6 7] A recent meta-analysis of randomised trials (ranging in sample size from 25 to 198 patients) suggest improvements in exercise capacity and HRQoL following intervention compared with control.[7] However, these previous studies have predominantly been supervised and delivered in centre-based settings. Participation in centre-based CR has been suboptimal, with national practice surveys indicating that fewer than 20% of eligible patients with HF may be receiving exercise-based CR.[8] Therefore,

**Table 4**  Caregiver outcomes at baseline and follow-up

| | Baseline | | Four-month follow-up | | Six-month follow-up | | Mean between-group difference* (95% CI) |
|---|---|---|---|---|---|---|---|
| | Intervention mean (SD), N | Control, mean (SD), N | Intervention, mean (SD), N | Control, mean (SD), N | Intervention, mean (SD), N | Control, mean (SD), N | |
| HADS, Anxiety | 8.6 (5.4), 10 | 6.2 (5.5), 10 | 7.1 (7.0), 8 | 6.8 (3.0), 10 | 6.3 (6.2), 8 | 7.6 (4.7), 9 | −3.4 (−6.6 to 0.2) |
| HADS, Depression | 4.0 (4.0), 10 | 4.7 (4.3), 10 | 3.9 (3.2), 8 | 5.4 (3.8), 10 | 2.9 (3.4), 8 | 5.9 (3.4), 9 | −2.3 (−5.1 to −0.5) |
| FAMQOL, Overall | 61.4 (10.5), 10 | 56.9 (12.0), 10 | 60.0 (10.2), 8 | 54.3 (12.6), 10 | 56.8 (8.6), 8 | 54.0 (8.7), 9 | −1.1 (−7.9 to 5.6) |
| FAMQOL, Physical | 17.0 (2.6), 10 | 14.9 (3.3), 10 | 15.9 (2.9), 8 | 14.9 (3.1), 10 | 15.8 (1.8), 8 | 15.0 (2.2), 9 | −1.2 (−2.7 to 0.3) |
| FAMQOL, Psychological | 13.9 (5.3), 10 | 13.5 (5.2), 10 | 13.3 (4.5), 8 | 12.0 (4.2), 10 | 12.8 (5.0), 8 | 12.1 (3.6), 9 | 0.3 (−2.7 to 3.3) |
| FAMQOL, Social | 16.6 (2.8), 10 | 15.8 (4.7), 10 | 16.3 (2.4), 8 | 14.8 (3.6), 10 | 15.6 (0.9), 8 | 14.8 (2.5), 9 | 0.0 (−1.6 to 1.5) |
| EQ-5D-3L, utility score | 0.78 (0.19), 10 | 0.74 (0.28), 10 | 0.81 (0.10), 8 | 0.75 (0.17), 10 | 0.77 (0.18), 8 | 0.67 (0.35), 9 | 0.07 (−0.08 to 0.22) |
| CBQ-HF, Physical | 4.5 (5.9), 10 | 3.7 (4.7), 10 | 2.0 (4.1), 8 | 6.3 (6.0), 10 | 4.4 (7.3) 8 | 5.2 (5.8), 9 | −1.5 (−4.1 to 1.1) |
| CBQ-HF, Emotional | 17.4 (13.8), 10 | 18.8 (13.0), 10 | 15.1 (13.3), 8 | 20.3 (12.0), 10 | 15.4 (16.0), 8 | 22.3 (13.1), 9 | −5.1 (−12.5 to 2.3) |
| CBQ-HF, Social life | 0.7 (1.2), 10 | 1.6 (2.0), 10 | 0.4 (0.7), 8 | 1.8 (2.1), 10 | 0.6 (1.1), 8 | 2.2 (2.5), 9 | −0.8 (−2.6 to 1.1) |
| CBQ-HF, Lifestyle | 1.9 (2.3), 10 | 4.3 (3.2), 10 | 2.5 (3.1), 8 | 4.4 (3.3), 10 | 2.4 (3.2), 8 | 6.0 (4.5), 9 | −1.4 (−4.7 to 1.9) |
| CC-SCHFI, Maintenance | 22.0 (11.0), 10 | 30.3 (15.7), 10 | 34.2 (25.1), 8 | 31.7 (14.6), 10 | 36.3 (23.5), 8 | 40.7 (17.9), 9 | 1.5 (−19.1 to 22.2) |
| CC-SCHFI, Management | 29.0 (21.6), 10 | 35.6 (14.7), 8 | 39.3 (28.2), 7 | 35.0 (18.5), 7 | 45.0 (13.2), 3 | 35.0 (19.1), 8 | 7.4 (−21.4 to 36.2) |
| CC-SCHFI, Confidence | 33.9 (15.6), 10 | 29.6 (19.8), 10 | 35.4 (17.6), 8 | 20.0 (17.2), 10 | 38.2 (16.4), 8 | 33.3 (20.6), 9 | 2.6 (−16.0 to 21.2) |

*Mean between-group differences (intervention minus control) adjusted for baseline values.
CBQ-HF, Caregiver Burden Questionnaire for Heart Failure; CC-SCHFI, Caregiver Contribution to SCHFI Questionnaire; FAMQOL, Family Caregiver Quality of Life Scale Questionnaire; HADS, Hospital Anxiety and Depression Scale; SCHFI, Self-care of Heart Failure Index

there is increasing interest in home-based programmes that have the potential to overcome these suboptimal rates of CR participation seen with HF.[10 11] To our knowledge, REACH-HF is the first comprehensive self-management CR intervention for patients with HFpEF and their caregivers that is home-based, with facilitation by healthcare professionals and whose development is informed by evidence, theory and input from service users—patients and clinicians.

The mechanism by which CR improves HRQoL in HFpEF remains unclear.[32] While exercise training has been shown to improve cardiac (systolic and diastolic) function in patients with HFrEF, studies have failed to show such consistent benefits in HFpEF.[6 7] Instead exercise training may improve exercise tolerance in HFpEF through peripheral mechanisms leading to an improved oxygen extraction in the active skeletal muscles.[33] Such improvements are likely to improve patient physical capacity and hence the physical component of HRQoL. Poor mental health, including depression in patients with HF is common and may be under-recognised and undertreated in cardiac populations such as HFpEF. This is supported by the baseline HADS Scores in this study indicating mild to moderate symptoms of depression and anxiety in a proportion of patients (and caregivers). A recent Cochrane review has shown comprehensive CR, including elements of stress management and exercise training, can have significant positive effects in terms in reductions in depression and anxiety of myocardial infarction and postrevascularisation populations.[34] The observed trend towards a reduction in depression and anxiety scores with the REACH-HF intervention points towards a possible basis of improvement in the mental component of HRQoL.

This study has a number of limitations. First, the study was not designed or powered to definitively assess the efficacy or safety of the REACH-HF intervention in HFpEF. Second, generalisability of this study's findings is limited, given it was based in a single centre. Thirdly, there was evidence of imbalances between intervention and control groups in their demographic characteristics and outcome scores at baseline. Fourthly, patient and clinician blinding was not possible in this study because of the nature of the intervention, although we did demonstrate that it was possible to blind outcome assessors to group allocation. Finally, the open label design of the study may have resulted in improvements in patient-reported outcomes in intervention participants as the result of placebo effects. However, we would note there was some evidence of fewer clinical events in the intervention group at 6 months. Given these limitations and the pilot nature of this trial, our findings should therefore be considered preliminary, and encouraging trends require confirmation in a larger, adequately powered clinical trial.

## Implications for planning a future trial

Based on the Minnesota Living With Heart Failure Questionnaire (MLWHFQ) total score as the primary outcome, a full trial comparing the REACH-HF plus usual care versus usual care alone would require recruitment of 210 patients with HFpEF per group. This estimate is based on detecting a minimum clinically important difference on the MLWHFQ of 5 points,[20] a SD of 25 points (as seen in this pilot trial, see table 1), a within-patient correlation of 0.8 (between baseline and the 6-month follow-up calculated from data from this pilot) and an assumed attrition rate of 10% (as seen in this pilot trial, see figure 1), at 90% power and 5% α level.

Two issues raised in this pilot that deserve consideration for a full trial include the choice of exercise test and the assessment of patient adherence to the REACH-HF intervention. In interviews, a number of patients in this study expressed the opinion that they found undertaking the ISWT an unpleasant experience; 12 of 45 (27%) patients were not able to undertake ISWT at the 6-month follow-up. This loss to follow-up my have resulted in bias in our assessment of exercise capacity over time and in our comparison of groups. Assessing and ensuring adequate levels of intervention adherence is a challenge in self-directed home-based interventions, such as REACH-HF.[11] Levels of patient attendance at face-to-face or telephone contacts with healthcare facilitators indicated good levels of intervention adherence. Patients were also asked to document changes in their health behaviours in a 'Patient Tracker' diary over the duration of the study. We need to examine if these diaries support our conclusion of good intervention adherence seen from facilitator contacts. It will be important to revisit these two issues in the design and planning of a future full trial.

In summary, the findings from this pilot study indicate that the REACH-HF home-based comprehensive self-management CR intervention facilitated by healthcare professionals is feasible, acceptable and suggests promising effects on patients with HFpEF and caregiver outcomes. This pilot study will help inform the funding application for a fully powered multicentre randomised trial to assess the clinical effectiveness and cost-effectiveness of the novel REACH-HF intervention in patients with HFpEF and their caregivers.

## PERSPECTIVES
### Competency in medical knowledge
The present findings support that patients with HFpEF have a substantial burden with exercise intolerance and a poor HRQoL.

**Author affiliations**
[1]School of Medicine, University of Dundee, Ninewells Hospital and Medical School, Dundee, UK
[2]School of Nursing and Health Sciences, University of Dundee, Dundee, UK
[3]Institute of Health Research, University of Exeter Medical School, Exeter, UK
[4]Research, Development and Innovation, Royal Cornwall Hospitals NHS Trust, Truro, UK
[5]Re-Cognition Health, Plymouth, UK

[6]Institute for Applied Health Research, University of Birmingham, Birmingham, UK
[7]Cardiology Department, Sandwell and West Birmingham Hospitals NHS Trust, Birmingham, UK
[8]Department of Health Sciences, University of York, York, UK
[9]Research and Development, Anuerin Bevan University Health Board, St Woolos Hospital, Newport, UK
[10]Cardiology Department, Royal Cornwall Hospitals NHS Trust, Truro, UK
[11]Centre for Exercise and Rehabilitation Science, University Hospitals of Leicester NHS Trust, Glenfield Hospital, Leicester, UK
[12]REACH-HF Patient and Public Involvement Group, c/o Research, Development & Innovation, Royal Cornwall Hospitals NHS Trust, Truro, UK
[13]Sport and Health Sciences, University of Exeter, Exeter, UK
[14]Peninsula Clinical Trials Unit, University of Plymouth, Plymouth, UK

**Acknowledgements** The authors thank the research nurses (Lynn Rutherford and Fatima Baig) intervention facilitators (Anona Cranston and Gillian Smith) and patients and their caregivers in Dundee and Tayside who participated in this study. The authors also thank the data team at Peninsula Clinical Trials Unit (University of Plymouth, UK), Lisa Price (University of Exeter, UK) who assisted with the analysis of accelerometry data, Karen Coyle (University of Dundee) who assisted with qualitative interviews and intervention fidelity checking, and Louise Taylor and her team at the Heart Manual Office Heart Office (Astley Ainslie Hospital, Edinburgh) for their assistance with the preparation and development of the REACH-HF intervention.

**Contributors** The REACH-HFpEF pilot trial was designed by CCL, KS, HMD, RST, JW, KJ, RCD, PJD, JM, RVL, SJS, CA, NB, CJG, CG, KP, MH, SS and CH. KJ, RST, RCD and HMD developed the original idea for REACH-HFpEF. FCW and CG analysed the data. VE and CH were responsible for study and data collection management. CCL undertook the first draft of the manuscript that was then edited by JW, FCW, RST and HMD. All authors were involved in critical evaluation and revision of the manuscript and have given final approval of the manuscript accepting responsibility for all aspects.

**Funding** This paper presents independent research funded by the National Institute for Health Research (NIHR) under its Programme Grants for Applied Research Programme (Grant Reference Number RP-PG-1210-12004). NB, CA, CJG and RST are also supported by the National Institute for Health Research (NIHR) Collaboration for Leadership in Applied Health Research and Care (CLAHRC) South West Peninsula at the Royal Devon and Exeter NHS Foundation Trust; KJ by CLAHRC West Midlands and SS by CLAHRC East-Midlands.

**Disclaimer** The views expressed are those of the authors and not necessarily those of the NHS, the NIHR or the Department of Health.

**Competing interests** RST is the lead for the ongoing portfolio of Cochrane reviews of cardiac rehabilitation. RST and HMD are named Scientific Advisors for the ongoing National Institute of Health and Care Excellence (NICE) updated clinical guidelines for the management heart failure (CG108). HMD is an ordinary member of the British Association for Cardiovascular Prevention and Rehabilitation (BACPR) council. All other coauthors declare no conflict of interest.

**Patient consent** Obtained.

**Ethics approval** Scotland A Research Ethics Committee (ISRCTN57596739).

**Provenance and peer review** Not commissioned; externally peer reviewed.

**Data sharing statement** The authors confirm that all data underlying the findings are fully available without restriction. The authors have made the clinical and economic data set available through the University of Exeter's Institutional Repository, Open Research Exeter (see https://ore.exeter.ac.uk). Access to these data is permitted but controlled through requests made via the repository to the chief investigator (Professor Taylor: r.s.taylor @exeter.ac.uk). Although use is permitted, this will be on the basis that the source of the data is acknowledged (including the funder) and it includes reference to the data set 'handle'.

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
