## [Reviewer comments · BMJ Open]

ARTICLE DETAILS

TITLE (PROVISIONAL)	A Randomised Controlled Trial of a Facilitated Home-Based Rehabilitation Intervention in Patients with Heart Failure with Preserved Ejection Fraction and their Caregivers: REACH-HFpEF Pilot Study.
AUTHORS	Lang, Chim C.; Smith, Karen; Wingham, Jennifer; Eyre, Victoria; Greaves, Colin; Warren, Fiona; Green, Colin; Jolly, Kate; Davis, Russell; Doherty, Patrick Joseph; Miles, Jackie; Britten, Nicky; Abraham, Charles; Van Lingen, Robin; Singh, Sally; Paul, Kevin; Hillsdon, Melvyn; Sadler, Susannah; Hayward, Christopher; Dalal, Hayes; Taylor, Rod

VERSION 1 – REVIEW

REVIEWER	Lene Theil Skovgaard Department of Biostatistics Institute of Public Health University of Copenhagen Denmark
REVIEW RETURNED	09-Oct-2017

GENERAL COMMENTS	How was the correction for baseline performed? Many of the outcomes are highly skewed, as seen from the large SD compared to the mean value. In these situations, SD cannot be meaningfully interpreted. The baseline corrected comparisons could be ok, but residuals should be investigated, and if skewness persists or variance is not homogeneous, the differences might instead be expressed in percentages (obtained by using logarithmic transformation of the outcomes), and transforming back afterwards. There are many tables, could some of them be transformed to figures? This would facilitate the overall picture of whether or not there was something gained by the intervention. If differences between groups were expressed as percentages, they could even be compared between the various outcomes. There is very little statistics in this paper, and I have not gone through all 94 pages, because it is beyond my field of expertise.
---

REVIEWER	John D Horowitz University of Adelaide, Australia
REVIEW RETURNED	27-Dec-2017

GENERAL COMMENTS	The issue addressed by this pilot study is the feasibility of undertaking a home-based cardiac rehabilitation program for patients with HFpEF. It is indeed shown that this is a feasible undertaking. What I believe would have been a more appropriate question is whether it is feasible to compare the results of home-based with hospital-based cardiac rehabilitation for such patients. Therefore I have two key questions for the investigators: (1) Surely the whole idea of attempting an intervention programme should be based on intention-to-treat? In other words, there may be more patients unwilling to participate in Programme A than in Programme B: this should be taken into account in the comparative evaluation. (2) What exactly was Usual Care? If there was no routine hospital-based rehabilitation, could the disproportionate activity associated with the home-based programme have resulted in some placebo effect, reflected in quality of life scores? More minor issues are: (1) How convincing is the evidence that cardiac rehabilitation is anything more than a placebo for patients with HFpEF? Is there any evidence that any "hard" parameter (eg BNP concentrations) is improved? (2) How was HFpEF diagnosed in this cohort? For example, was VO₂max determined? (3) Patients were assessed after 3 and 6 months. Should not both these time points have been taken into account in overall assessment, for example via an ANOVA with repeated measures design?
---

VERSION 1 – AUTHOR RESPONSE

Editorial Request:

1.1 Please replace the CONSORT checklist for randomised trials with the CONSORT extension for pilot and feasibility trials.

Authors' response:

1.1 We have replaced the CONSORT checklist with the CONSORT extension for pilot and feasibility trials as requested. We have attached the new CONSORT checklist.

BMJ Open REVIEWER COMMENTS:

Reviewer: 1

Reviewer Name: Lene Theil Skovgaard

Institution and Country: Department of Biostatistics, Institute of Public Health, University of Copenhagen, Denmark

Competing Interests: None declared

Reviewer's Comments:

1.1. How was the correction for baseline performed?

Authors' response:

1.1 All between group analyses for outcomes measured at 6-month follow-up were performed using a linear regression model with inclusion of baseline score for the relevant variable. We have amended the text to clarify this.

Reviewer's Comments:

1.2 Many of the outcomes are highly skewed, as seen from the large SD compared to the mean value. In these situations, SD cannot be meaningfully interpreted.

The baseline corrected comparisons could be ok, but residuals should be investigated, and if skewness persists or variance is not homogeneous, the differences might instead be expressed in percentages (obtained by using logarithmic transformation of the outcomes), and transforming back afterwards.

Authors' Comments:

1.2 We agree that several of the outcomes are skewed and that in this instance a linear model may be inappropriate for a between group comparison, due to the small sample size, which is not robust to such skewness. As this is a pilot study, we do not have sufficient power to form conclusions regarding treatment effect based on an inferential analysis. While the use of a transformation, such as a logarithmic transformation as suggested, may result in less skewness and be more appropriate to a regression model, we feel that reporting the outcomes as percentages would be less helpful to clinicians as these outcomes are usually reported using the actual scores.

Our purpose in reporting the between group mean difference and its confidence interval is to indicate the direction of the observed difference and to provide a range of values in which the treatment effect for a fully powered trial may be expected to lie, rather than to use the confidence interval to make an inferential comparison, and we do not report p-values from these analyses. In a pilot study such as this it is a matter for debate whether it is desirable to report between group comparisons rather than simply reporting outcomes descriptively, although we have reported between group mean differences and confidence intervals in other similar studies. We would prefer to retain the mean difference as it clearly indicates the magnitude and direction of the treatment effect, and although the CI may be biased towards being wider than would be the case using a transformation, this is not a major drawback as the CI should be interpreted in the context of providing a sense of the range of potential treatment effects that may be observed in a full trial, rather than in the context of trying to draw inferential conclusions.

Reviewer's Comments:

1.3 There are many tables, could some of them be transformed to figures? This would facilitate the overall picture of whether or not there was something gained by the intervention. If differences between groups were expressed as percentages, they could even be compared between the various outcomes.

Authors' response:

1.3 We agree that there are a large number of reported outcomes in the tables. Accordingly, we have transformed the Primary endpoint of MLWHF scores into a new Figure 2.

However, we feel that we should retain the reported numerical outcomes in the table to indicate the direction of treatment effects and would not wish to replace the descriptive means and standard deviations as these may be useful to other researchers in future. We will ensure that the directionality of scales is defined in Appendix 2 and will revise the formatting of the tables to indicate clearly where the outcome is more positive for the intervention group.

Reviewer's Comments:

1.4 There is very little statistics in this paper, and I have not gone through all 94 pages, because it is beyond my field of expertise.

Authors' response:

1.4 Further details related to this study can be found in our rationale and protocol paper that was previously published in BMJ Open (Eyre V, Lang CC, Smith K, et al Rehabilitation Enablement in Chronic Heart Failure—a facilitated self-care rehabilitation intervention in patients with heart failure with preserved ejection fraction (REACH-HFpEF) and their caregivers: rationale and protocol for a single-centre pilot randomised controlled trial. BMJ Open. 2016;6(10)). As above, we have included some additional details in the statistical analysis section.

Reviewer 2

Reviewer Name: John D Horowitz

Institution and Country: University of Adelaide, Australia

Competing Interests: None declared

Reviewer's Comments:

2.1 The issue addressed by this pilot study is the feasibility of undertaking a home-based cardiac rehabilitation program for patients with HFpEF. It is indeed shown that this is a feasible undertaking. What I believe would have been a more appropriate question is whether it is feasible to compare the results of home-based with hospital-based cardiac rehabilitation for such patients.

Authors' response:

2.1 We thank the reviewer for noting that we have shown that our pilot study have shown the feasibility of our home-based cardiac rehabilitation program for patients with HFpEF. We appreciate the reviewer raising the issue of considering a feasibility study to compare home-based with hospital-based cardiac rehabilitation for HFpEF patients. However, it is important to note that at this time, cardiac rehabilitation is not part of the usual care of patients with HFpEF. Clinical practice guidelines including those of NICE do not specifically recommend cardiac rehabilitation in patients with HFpEF. Hence the choice of no rehabilitation (usual care) comparator in this trial.

Reviewer's Comments:

2.2 Surely the whole idea of attempting an intervention programme should be based on intention-to-treat? In other words, there may be more patients unwilling to participate in Programme A than in Programme B: this should be taken into account in the comparative evaluation

Authors' response:

2.2 This was an intention to treat study i.e. outcomes were compared between groups according to the original random allocation. In our study, there were no 'drop outs' of patients who had been randomised to usual care.

Reviewer's Comments:

2.3 What exactly was Usual Care? If there was no routine hospital-based rehabilitation, could the disproportionate activity associated with the home-based programme have resulted in some placebo effect, reflected in quality of life scores?

Authors' Response:

2.3 Both intervention and control groups received the usual care of conventional medical management for HF according to current guidelines i.e., attendance and care from the GP / primary care and follow up by hospital physician / cardiologist after diagnosis of HFpEF. We agree with the reviewer that the open label design of the study may have resulted in some placebo effects and have added this point as a limitation.

Reviewer's Comments

2.4 More minor issues are:(1) How convincing is the evidence that cardiac rehabilitation is anything more than a placebo for patients with HFpEF? Is there any evidence that any "hard" parameter (eg BNP concentrations) is improved?

Authors' response:

2.4 As above, we have added placebo effects as a potential limitation of the trial. We did not collect BNP as secondary outcome. However, we did assess 'hard end points at 6 months; 11 (4 intervention, 7 control) patients experienced a hospital admission with 4 (all control) of these admissions being heart failure-related. Any benefit on clinical events will need to be confirmed in a larger multi-centre randomised outcome trial that is proposed and planned.

Reviewer's Comments

2.5 How was HFpEF diagnosed in this cohort? For example, was VO2max determined?

Author's response:

2.5 The diagnosis HFpEF is based on the following criteria that was detailed in our rationale and protocol paper (Eyre V, Lang CC, Smith K, et al Rehabilitation Enablement in Chronic Heart Failure—a facilitated self-care rehabilitation intervention in patients with heart failure with preserved ejection fraction (REACH-HFpEF) and their caregivers: rationale and protocol for a single-centre pilot randomised controlled trial. *BMJ Open*. 2016;6(10)). The details are shown below:

Trial entry criteria

Inclusion criteria

Patients with heart failure, defined by the presence of at least one of the following symptoms at the time of screening:

- paroxysmal nocturnal dyspnoea
- or orthopnoea
- or dyspnoea on mild or moderate exertion

AND at least one of the following signs prior to study entry:

- basal crepitations
- or elevated jugular venous pressure
- or lower extremity oedema
- or chest radiograph demonstrating pleural effusion, pulmonary congestion or cardiomegaly.

Patients with left ventricular ejection fraction (EF) $\geq 45\%$ obtained within 6 months prior to randomisation and after any myocardial infarction (MI) or other event that would affect EF (ideally obtained by echocardiography, although radionuclide ventriculography and angiography are acceptable).

Participating patients were aged 18 years or older and had a confirmed diagnosis of HFpEF on echocardiography, radionuclide ventriculography or angiography (i.e. left ventricular ejection fraction $\geq 45\%$ within the last 6 months prior to randomisation).

A submaximal exercise test (incremental shuttle walk test) was undertaken in this pilot trial to assess exercise capacity. We purposively did not seek cardiopulmonary exercise testing to directly assess VO2max to help diagnose HFpEF as this would have deterred patients from consenting to participate and importantly, cardiopulmonary exercise testing with gas exchange analysis is not widely available in the UK.

Please note we have corrected minor typographical errors and grammar which we noticed when we were making the above changes.

VERSION 2 – REVIEW

REVIEWER	John D Horowitz University of Adelaide, Australia
REVIEW RETURNED	27-Feb-2018

GENERAL COMMENTS	The background for this pilot study is that the feasibility of administering a home-based group rehabilitation programme for patients with heart failure plus a preserved left ventricular ejection fraction remains unexplored. The current investigation explores both feasibility and cost-effectiveness of such a programme, establishing that it could be performed without major limitations, and thus would be able to provide outcome parameters for this large subgroup of heart failure patients. The work involved here is a necessary component for the preparation of a large clinical study of management of heart failure with preserved ejection fraction.
--